# Optimal Evacuation Strategy for Parking Lots Considering the Dynamic Background Traffic Flows

**DOI:** 10.3390/ijerph16122194

**Published:** 2019-06-21

**Authors:** Xinhua Mao, Changwei Yuan, Jiahua Gan, Jibiao Zhou

**Affiliations:** 1School of Economics and Management, Chang’an University, Xi’an 710064, China; changwei@chd.edu.cn; 2Department of Civil and Environmental Engineering, University of Waterloo, Waterloo, ON N2L 3G1, Canada; 3Transport Planning and Research Institute, Ministry of Transport, Beijing 100028, China; ganjh@tpri.org.cn; 4School of Civil and Transportation Engineering, Ningbo University of Technology, Ningbo 315211, China; zhoujb2014@nbut.edu.cn; 5College of Transportation Engineering, Tongji University, Shanghai 201804, China

**Keywords:** optimal evacuation strategy, parking lot, dynamic background traffic flows, queuing theory, departure rate, average queuing time, average travel time, total evacuation time

## Abstract

An optimal evacuation strategy for parking lots can shorten evacuation times and reduce casualties and economic loss. However, the impact of dynamic background traffic flows in a road network on the evacuation plan is rarely taken into account in existing approaches. This research develops an optimal evacuation model with total evacuation time minimization by dividing the evacuation process in a parking lot into two periods. In the first period, a queuing theory is used to estimate the queuing time, and in the second period, a traffic flow equilibrium model and an intersection delay model are employed to simulate vehicles’ route choice. To deal with these models, a modified ant colony algorithm is developed. The results of a numerical example prove that the proposed method has an advantage in improving evacuation efficiency. The results also show that background traffic flows affect not only vehicles’ average queuing time in parking lots but also optimal evacuation route choice. Additionally, a sensitivity analysis indicates that the minimum threshold of headway time that allows vehicles out of a parking lot to merge into the background traffic flows on the roads connecting the exits has a great impact on average queuing time, average travel time, and total evacuation time.

## 1. Introduction

Vehicle evacuation refers to the guiding of vehicles gathered at a dangerous place to safe areas within an effective time by traffic control measures when emergencies happen [1]. As an important part of disaster mitigation and emergency disposal, an optimal evacuation management strategy must be made during the evacuation process that comprehensively considers road network capacities, traffic management equipment, emergency response resources, real-time traffic conditions, etc., to reduce casualties and economic losses [2]. Due to the increasing growth of vehicles and emergencies, the vehicle evacuation problem has attracted more and more attention from researchers [3].

The parking lot is a kind of common public infrastructure in cities where a lot of cars usually gather. Hence, an effective vehicle evacuation strategy for emergencies is necessary for the safety and security management of parking lots. To increase vehicles’ departure rate, a single parking lot is usually designed with more than one exit connecting multiple roads [4], which makes evacuation both a multi-origin and multi-destination problem. In the actual evacuation process of a parking lot, traffic capacity and delay on every road in the network change over time, and they are dependent on both background traffic flows and evacuation traffic flows. However, most traditional vehicle evacuation models ignore the dynamics of background traffic flows [5,6,7]. The impact of background traffic flows on evacuation strategy, including the queue of vehicles in a parking lot and the choice of optimal routes in the evacuation road network, has rarely been investigated, which should be taken into consideration in optimal vehicle evacuation modeling. Additionally, a comprehensive and accurate estimation of total evacuation time has not been completely investigated [8].

To fill this gap, we have divided the whole evacuation process for a parking lot into two periods. For the first period, we have developed a queuing model that considers the impact of background traffic flow on vehicles’ departure rate—thus allowing for the estimation of the average queuing time of vehicles in parking lots [9,10]. For the second period, we have employed a traffic flow equilibrium model and an intersection delay model to simulate optimal evacuation routes, which also helps calculate travel time and estimate delays at intersections under traffic signal control [11,12]. For the better generation of optimal evacuation strategies for parking lots, this research extends the general approach in three aspects: (i) It formulates the total evacuation time function including the queuing time, the travel time, and the delay at intersections based on the two-period simulation; (ii) it develops an optimal evacuation model for parking lots with total evacuation time minimization; and (iii) it proposes a modified ant colony algorithm as the model solution.

This research makes the following two contributions. Firstly, we develop a queuing model at a parking lot affected by the time headway of background traffic flow on the roads—thus allowing for the estimation of the average queuing time of vehicles in parking lots. Secondly, we propose a framework with minimum evacuation times to generate optimal vehicle evacuation strategies for parking lots, considering the dynamics of background traffic flows.

The remainder of this paper is organized as follows. Section 2 reviews the studies on vehicle evacuation modeling. Section 3 gives the assumptions, proposes the methodology framework in detail, and develops a modified ant colony algorithm as the model solution. Section 4 describes a numerical example and presents the simulation results. Section 5 makes a discussion on the key results obtained from the model and performs a sensitivity analysis. Section 6 draws the main conclusions.

## 2. Literature Review

Vehicle evacuation problems have attracted tremendous attention from researchers [13,14,15]. In the existing literature, a wide range of modeling approaches have been developed, and these can be classified into two categories: (i) The analytical approach and (ii) the simulation-based approach.

The analytical approach aims to obtain optimal evacuation plans including the shortest evacuation time, the best evacuation routes, and the optimal allocation of evacuees. For example, Sheffi et al. developed a macro model with minimum network clearance time which can simulate traffic patterns during an emergency evacuation [16]. To obtain the shortest evacuation time, Yamada described a minimal cost flow model that considers the traffic capacity limit of a road network [17]. However, these two models are based on static road networks which assume that traffic parameters, e.g., traffic density and traffic volume, are constant values that do not vary during an emergency evacuation process. Since traffic characteristics usually have time-dependent changes, the static road network-based approach has limitations in highly accurate simulations. To fill this gap, many other researchers developed evacuation models based on the dynamic road network which take the time into account and make the best of the performance of the entire network system. For instance, Cova and Johnson formulated an evacuation in a complex road network as an integer extension of the minimum-cost flow problem, which was solved by a mixed-integer programming model [18]. To obtain the optimal evacuation route in a hierarchical directed network, Fang et al. developed a multi-objective optimization model aiming at minimum evacuation time, evacuation distance, and congestion degrees [19]. Sbayti and Mahmassani proposed a modified system-optimal dynamic traffic assignment model with road network clearance time minimization to obtain evacuation trips; this model took the delays between origins and destinations into consideration [20]. Based on the above network optimization methods, some uncertainties were also captured in the evacuation models. For example, Bretschneider and Kimms put forward a mixed-integer evacuation model with a minimization of evacuation time which considered the uncertainties of conflicts within intersections [21]. Additionally, other researchers employed a cell-transmission model as an analytical approach to investigate the evacuation problem. Chiu and Zheng employed a cell transmission model-based linear-programming model with simultaneous mobilization strategies which addressed the evacuation plan in a network with different destinations [22]. Likewise, Tak et al. utilized an agent-based cell transmission model to deal with evacuation decisions on destinations and travel directions [23].

The simulation-based approach is another technique used to deal with evacuation problems using traffic assignment simulation models. Murray-Tuite and Mahmassani developed a two-stage evacuation model utilizing a micro assignment simulation procedure to mimic the resulting traffic interactions in a road network [24]. Lämmel et al. presented a robust and flexible simulation framework to predict the evacuation process in a large scale road network [25]. Balakrishna et al. also proposed an adaptive simulation framework for evacuation modeling in different emergent situations [26]. An effective optimization-based simulation procedure was defined by Kimms and Maassen to obtain the optimal routes during an emergency situation in urban areas [27].

Despite the wide range of evacuation modeling approaches, it is rare in literature to incorporate dynamic background traffic flow into evacuation strategy modeling. This research proposes an optimal vehicle evacuation model for parking lots using a road traffic flow equilibrium model and an intersection delay model combined with a queuing model to consider the interaction between dynamic background traffic flow and queuing time. Only in this way can the total time consumption during the evacuation process be evaluated accurately and comprehensively. Furthermore, a modified ant colony algorithm is developed to solve the model.

## 3. Materials and Methods

### 3.1. Assumptions

For simplicity, the following assumptions are given in this section.
(1)Each exit of the modelled parking lot connects only one road.(2)Each exit of the parking lot has one lane, which only serves one vehicle at the same time.(3)The evacuation process will not lead to traffic paralysis in the road network.(4)The vehicle traffic flows on the road network will not be disturbed by non-motor vehicles or pedestrians.(5)Vehicles evacuated from the parking lot enter into the road network only by a right turn.

### 3.2. Two Periods of the Evacuation Process in a Parking Lot

The vehicle evacuation process in the parking lot can be divided into two periods, as shown in Figure 1. 

In the first period—i.e., the time between when a vehicle is generated and when a vehicle is actually loaded onto the road—vehicles have to queue and wait for departure in the parking lot because the traffic capacity of the network usually cannot afford the rapid growth of the traffic need caused by an emergency evacuation. 

In the second period—i.e., the time between when vehicles merge into the traffic flows in the road network and when the vehicles reach their final destinations—traffic flows are constituted of background traffic flows and evacuation traffic flows, and the two kinds of traffic flows interact with each other. 

Hence, the background traffic flows in the network not only affect vehicles’ queuing time in the parking lot but also determine vehicles’ optimal evacuation route choice. In view of this, dynamic background traffic flows should be considered for the optimal evacuation strategy of the parking lot.

### 3.3. Queuing Modeling in the Parking Lot (the First Period)

Assume that there are *A* vehicles in the parking lot, which has *R* exits. When the evacuation begins, vehicles will randomly choose to queue at any exit waiting to leave, which generates *R* queues in the parking lot, shown in Figure 2. We define the queue at the exit *r* as queuing system *r*, *r* = 1,2,⋯,*R*.

According to this queuing theory, the queue at any single exit of the parking lot can be denoted as a M/M/1/1 queuing system [28]. M/M stands for Poisson arrivals and the negative, exponentially distributed lengths of stay [29]; 1/1 indicates that there is only 1 exit which can only serve 1 vehicle at a time [30]. The average queuing time (including waiting time and service time) of queuing system *r* is calculated as in [31].
(1)$${\bar{d}}_{r}=\frac{1}{\mu_{r}-\lambda_{r}},\forall r\in R$$
where ${\bar{d}}_{r}$ is the average queuing time of queuing system *r*; $\mu_{r}$ is the average arrival rate of queuing system *r*, which is the reciprocal of the average time between the arrivals of two consecutive vehicles; and $\lambda_{r}$ is the vehicle departure rate of queuing system *r*, which is the reciprocal of the service time.

However, for a parking lot, the service time $1/\lambda_{r}$ actually means the period from the time a vehicle arrives at the front of the queue until it merges into the traffic flow on the road connecting the exit *r*, a period determined by the background traffic flow. The derivation process of $1/\lambda_{r}$ is described in detail as follows.

The background traffic volume on the road connecting *r*th exit of the parking lot is represented by $Q_{r}$ (vehicle/s); the minimum threshold of time headway (interval between two consecutive vehicles) of the background traffic allowing the vehicles out of the exit *r* to merge into the background traffic is $\tau_{r}$. Hence, the possibility that a single vehicle can merge into the background traffic flow, i.e., the possibility that the time headway h of the background traffic flow is more than $\tau_{r}$, is formulated as in [32].
(2)$$P_{r}(h>\tau_{r}{)=e}^{{-Q}_{r}\cdot \tau_{r}},\forall r\in R$$
where $P_{r}(h\;>\;\tau_{r})$ is the possibility that the time headway h on the road connecting *r*th exit of the parking lot is more than $\tau_{i}$.

Accordingly, during the time period *t*, the number of intervals (${h\;>\;\tau }_{r}$) of the background traffic flow is calculated as.
(3)$$N_{\mathrm{interval}}(h>\tau_{r}{)=Q}_{r}\cdot t\cdot e^{{-Q}_{r}\cdot \tau_{r}},\forall r\in R$$

The number of vehicles between two intervals is
(4)$$N_{\mathrm{vehicle}}=\frac{Q_{r}\cdot t}{Q_{r}\cdot t\cdot e^{{-Q}_{r}\cdot \tau_{r}}}{=e}^{Q_{r}\cdot \tau_{r}},\forall r\in R$$

The average time between two consecutive intervals is formulated as
(5)$$T_{r}{=(e}^{Q_{r}\cdot \tau_{r}}-1)\times (\frac{1}{Q_{r}}-\frac{e^{{-Q}_{r}\cdot \tau_{r}}}{{1-e}^{{-Q}_{r}\cdot \tau_{r}}})=\frac{{1-e}^{{-Q}_{r}\cdot \tau_{r}}}{e^{{-Q}_{r}\cdot \tau_{r}}}\times (\frac{1}{Q_{r}}-\frac{e^{{-Q}_{r}\cdot \tau_{r}}}{{1-e}^{{-Q}_{r}\cdot \tau_{r}}})=\frac{1}{Q_{r}\cdot e^{{-Q}_{r}\cdot \tau_{r}}}-\frac{1}{Q_{r}}{-\tau }_{r},\forall r\in R$$

Since the vehicle at the front of the queuing system *r* can only leave and merge into the background traffic flow when $h\;>\;\tau_{r}$, the average time between the two intervals can be denoted as the service time of the queuing system *r*.
(6)$$\frac{1}{\lambda_{r}}{=T}_{r}=\frac{1}{Q_{r}\cdot e^{{-Q}_{r}\cdot \tau_{r}}}-\frac{1}{Q_{r}}{-\tau }_{r},\forall r\in R$$

Accordingly, Equation (1) can be rewritten as Equation (7).
(7)$${\bar{d}}_{r}=\frac{1}{\mu_{r}-\frac{1}{T_{r}}}=\frac{T_{r}}{\mu_{r}\cdot T_{r}-1}=\frac{e^{Q_{r}\cdot \tau_{r}}-Q_{r}\cdot \tau_{r}-1}{\mu_{r}\cdot (e^{Q_{r}\cdot \tau_{r}}-Q_{r}\cdot \tau_{r}{-1)-Q}_{r}},\forall r\in R$$

As a result, the sum of all vehicles’ queuing time in the parking lot is calculated as
(8)$$T_{Q}=\overset{R}{\underset{r=1}{\sum }}A_{r}\cdot {\bar{d}}_{r}$$
where $T_{Q}$ is the sum of all vehicles’ queuing time in the parking lot and $A_{r}$ is the number of vehicles in the queuing system *r*, which should be subject to
(9)$$\overset{R}{\underset{r=1}{\sum }}A_{r}=A$$

### 3.4. Network Traffic Flow Modeling (the Second Period)

The second period of the evacuation process for a parking lot with multiple exits can be described as a multi-origin and multi-destination evacuation problem [33]. We represent a multi-origin and multi-destination evacuation network in Figure 3, where vehicles are evacuated from origins *V*_1_ through mid-points *V*_2_ to destinations *V*_3_.

#### 3.4.1. Traffic Flow Equilibrium Modeling on Nodes

Inspired by the network traffic flow theory, for every mid-point in an evacuation network, equilibrium can be achieved between the total inflow traffic (including background traffic and evacuation traffic) and the outflow traffic [34]. This is formulated as
(10)$$\overset{N}{\underset{i=1}{\sum }}\left(q_{ij}{+q}_{ij}^{*}\right)-\overset{N}{\underset{i=1}{\sum }}\left(q_{ji}{+q}_{ji}^{*}\right)=0,\;\forall i,j\in V_{2}$$

Additionally, the total evacuation traffic generated by the origins equals the total evacuation traffic attracted by destinations. This is described as
(11)$$\overset{M}{\underset{s=1}{\sum }}q_{sj}-\overset{U}{\underset{g=1}{\sum }}q_{jg}=0,\;\forall j\in V_{2},\;\forall s\in V_{1},\;\forall g\in V_{3}$$

#### 3.4.2. Evacuation Traffic Equilibrium Modeling on Links

According to the trip distribution theory, the traffic flow distribution in the network is determined by the traffic demands between the origins and destinations [35]. Hence, the evacuation traffic equilibrium is formulated as
(12)$$\underset{k\in K_{sg}}{\sum }h_{k}^{sg}{=Q}_{sg},\;\forall s\in V_{1},\;\forall g\in V_{3}$$
(13)$$\underset{k\in K_{sg}}{\sum }\underset{s\in V_{1}}{\sum }\underset{g\in V_{3}}{\sum }h_{k}^{sg}\cdot \delta_{ij}^{sg,k}{=q}_{ij},\forall i,\;j\in V_{1}\cup V_{2}\cup V_{3}$$
where Equation (12) is the evacuation traffic conservation which indicates the evacuation demands from origin *s* to destination *g* is the sum of evacuation traffic flows on all routes connecting origin *s* to destination *g* [36]. Equation (13) indicates that the evacuation traffic flow on every single link is the sum of evacuation traffic flows on all routes on which the link lies [37].

#### 3.4.3. Travel Time on Links

The travel time on links is affected by actual traffic flow. We use Equation (14), as in [38], to establish the travel time function related to traffic flows.
(14)$$t_{ij}{(q}_{ij}{+q}_{ij}^{*})=\frac{l_{ij}}{v_{ij0}}\cdot \left(1+\psi \cdot {\left(\frac{q_{ij}{+q}_{ij}^{*}}{c_{ij}}\right)}^{\xi }\right)$$

Hence, the total travel time of all vehicles evacuated is calculated as
(15)$$T_{T}=\overset{N}{\underset{i,\;j}{\sum }}\underset{k\in K_{sg}}{\sum }\delta_{ij}^{sg,k}\cdot \int_{0}^{q_{ij}}t_{ij}\left(x\right)dx,\;\forall i,\;j\in V_{1}\cup V_{2}\cup V_{3}$$

### 3.5. Simulation of Intersection Delay under Signal Control

We use the approach as in [39] to simulate the intersection delay under signal control. This is formulated as.
(16)D¯ij=μ(1−λ)22(1−λy)+y22(qij+qij*)·(1−y)−ω·(μ(qij+qij*)2)13·y(2+λ5)
where ${\bar{D}}_{ij}$ is the average delay at the intersection *j* on the route (*i*, *j*), μ is the signal cycle time, λ is the green time ratio, *y* is the degree of saturation, and ω is a field calibration coefficient.

Hence, the total intersection delays $T_{D}$ of all vehicles on their evacuation routes are calculated as
(17)$$T_{D}=\overset{N}{\underset{i,j}{\sum }}\underset{k\in K_{sg}}{\sum }\delta_{ij}^{sg,k}\cdot q_{ij}\cdot {\bar{D}}_{ij},\forall i,j\in V_{1}\cup V_{2}\cup V_{3}$$

### 3.6. Optimal Evacuation Modeling


(18)$${\mathrm{Minimize}\;T=T}_{Q}{+T}_{T}{+T}_{D}$$


Subject to the constraints
(19)$$\overset{N}{\underset{i=1}{\sum }}\left(q_{ij}{+q}_{ij}^{*}\right)-\overset{N}{\underset{i=1}{\sum }}\left(q_{ji}{+q}_{ji}^{*}\right)=0,\;\forall i,\;j\in V_{2}$$
(20)$$\overset{M}{\underset{s=1}{\sum }}q_{sj}-\overset{U}{\underset{g=1}{\sum }}q_{jg}=0,\;\forall j\in V_{2},\;\forall s\in V_{1},\;\forall g\in V_{3}$$
(21)$$\underset{k\in K_{sg}}{\sum }h_{k}^{sg}{=Q}_{sg},\;\forall s\in V_{1},\;\forall g\in V_{3}$$
(22)$$\underset{k\in K_{sg}}{\sum }\underset{s\in V_{1}}{\sum }\underset{g\in V_{3}}{\sum }h_{k}^{sg}\cdot \delta_{ij}^{sg,k}{=q}_{ij},\forall i,\;j\in V_{1}\cup V_{2}\cup V_{3}$$
(23)$$\underset{s\in V_{1}}{\sum }\underset{g\in V_{3}}{\sum }Q_{sg}=A\;$$
(24)$$0\leq q_{ij}{+q}_{ij}^{*}\leq c_{ij},\forall i,\;j\in V_{1}\cup V_{2}\cup V_{3}$$
(25)$$h_{k}^{sg}\geq 0,\forall k\in K_{\mathrm{st}}$$
where Equation (18) is the objective function which ensures total evacuation time minimization including queuing time, travel time, and intersection delay; Equations (19) and (20) are traffic flow equilibrium constraints on nodes; Equations (21) and (22) are traffic flow equilibrium constraints on links; Equation (23) means that the sum of the evacuation demands from all origins to destinations is equal to the vehicles in the parking lot; Equation (23) is the traffic capacity constraint; and Equation (24) makes sure that the traffic flow on every single route is nonnegative.

### 3.7. Model Solution

An ant colony algorithm is a common method to solve optimization problems which has advantages such as a strong robustness and a great global searching ability [40,41]. However, it has a slow convergence rate and a poor local searching ability [42,43]. Hence, we developed a modified ant colony algorithm to solve the optimal evacuation model proposed above. The modification was made to focus on the strategy of pheromone updating and the formulation of a heuristic function. The detailed steps of the algorithm are as follows.

Step 1: Initialization. Denote the initial number of iterations $N_{\mathrm{iteration}}\;$= 0; set the initial value of the pheromone on every link to $\vartheta_{ij}\left(0\right)$ and $\Delta \vartheta_{ij}\;$= 0; and set the number of ants to *m*.

Step 2: Start the iteration. Set $N_{\mathrm{iteration}}=N_{\mathrm{iteration}}\;+\;1$. Ant k (k=1,2,⋯,m) chooses its route based on the pheromone of each link. The possibility for ant *k* to determine the direction of its transition from node *i* to node *j* at time *t* is formulated as
(26)$$P_{ij}^{k}(t)=\{\begin{array}{cc} \frac{\vartheta_{ij}^{\alpha }\left(t\right)\cdot \eta_{ij}^{\beta }\left(t\right)}{\sum_{p\in {\mathrm{allowed}}_{k}}\vartheta_{ip}^{\alpha }\left(t\right)\cdot \eta_{ip}^{\beta }\left(t\right)} & j\in {\mathrm{allowed}}_{k}\\ \;0\; & \mathrm{otherwise} \end{array}$$
(27)$$\eta_{ij}\left(t\right)=\frac{1}{t_{ij}+{\bar{D}}_{ij}}$$
where $P_{ij}^{k}\left(t\right)$ is the possibility for ant *k* to determine the direction of its transition from node *i* to node *j* at time *t*; $\vartheta_{ij}\left(t\right)$ is the value of the pheromone on link (*i*, *j*) at time *t*; $\eta_{ij}\left(t\right)$ is a heuristic function on link (*i*, *j*) which is the reciprocal of the sum of travel time $t_{ij}$ and intersection delay ${\bar{D}}_{ij}$ on link (*i*, *j*); ${\mathrm{allowed}}_{k}$ is a dynamic node set which represents the nodes that ant *k* can choose for its next transition; *p* is the node that ant *k* can choose for its next transition, $p\in {\mathrm{allowed}}_{k}$; and α is a heuristic information factor which is defined as the role of ant's accumulative information in its route selection. The higher the value of α is, the more inclined the ant is to choose the route passed by other ants. Finally, β is the heuristic factor of expectation which indicates the importance of heuristic information in ant's route selection. The greater the value of β is, the higher the possibility of the transition is.

Additionally, a tabu table ${\mathrm{tabu}}_{k}$ is created to record the routes of ant *k*. 

Step 3: If every single ant finds a feasible route, the shortest path in time is identified as the optimal route obtained by this iteration. Otherwise, return to Step 2 and execute the procedure until all ants find their feasible routes.

Step 4: Pheromone update. The pheromone on every link is updated after every ant finishes its ergodic operation. In this research, we updated the pheromone according to the following principles.
(28)$$\vartheta_{ij}\left(t+1\right)=\left(1-\rho \right)\cdot \vartheta_{ij}\left(t\right)+\Delta \vartheta_{ij}\left(t\right)$$
(29)$$\Delta \vartheta_{ij}\left(t\right)=\overset{m}{\underset{k=1}{\sum }}\Delta \vartheta_{ij}^{k}\left(t\right)$$
where ρ is the pheromone volatilization coefficient which belongs to (0, 1), $\Delta \vartheta_{ij}\left(t\right)$ is the pheromone change on the link (*i*, *j*), and $\Delta \vartheta_{ij}^{k}\left(t\right)$ is the pheromone left on the link (*i*, *j*) by the *k*th ant. $\Delta \vartheta_{ij}^{k}\left(t\right)$ is calculated by Equation (30).
(30)$$\Delta \vartheta_{ij}^{k}(t)=\{\begin{array}{cc} \eta_{ij}^{k}\left(t\right) & \;\mathrm{If}\;\mathrm{the}\;k\mathrm{th}\;\mathrm{ant}\;\mathrm{goes}\;\mathrm{along}\;\mathrm{the}\;\mathrm{route}\;(i,\;j)\\ 0\; & \;\mathrm{Otherwise}\; \end{array}$$
where $\eta_{ij}^{k}\left(t\right)$ is the reciprocal of the sum of the *k*th ant’s travel time and intersection delay on link (*i*, *j*).

Step 5: Repeat the procedure from steps 2 to 4 until $N_{\mathrm{iteration}}$ iterations are completed.

## 4. Numerical Example

A city parking lot in Xi’an, a city of western China, was employed to test the approach framework proposed in this research. The layout of the road network is plotted in Figure 4. We assumed that a maintenance action would be conducted to the parking lot at 2:00 p.m. on Friday when there would be 860 vehicles in the parking lot. In order to prevent serious traffic congestion in the road network, these vehicles would have to be evacuated two kilometers away from the parking lot, i.e., the vehicles would have to be evacuated to the area other than a circle with a diameter of two kilometers. The area inside the circle was defined as a dangerous area, while the area outside the circle was the safe area.

In this numerical example, four exits of the parking lot $V_{1}\;=\;\left\{E1,E2,E3,E4\right\}$ were denoted as the set of origins of evacuation routes; $V_{2}\;=\;\left\{C1,C2,\cdots ,C17\right\}$ were denoted as the set of mid-points of evacuation routes; and $V_{3}\;=\;\left\{A1,A2,\cdots ,A15\right\}$ were defined as the set of destinations of evacuation routes. 

### 4.1. Data Collection

The network constituted nine roads and 17 intersections. We collected the length, traffic capacity, the free flow speed of each road segment. The traffic volumes on all road segments between two intersections were also collected every five minutes from 13:00 to 14:00 and transferred back to the laboratory in real time. The data of the road network and traffic volumes are shown in Appendix A, Table A1.

The 17 intersections were all controlled by traffic signal systems. The signal cycle time and green time ratio of each traffic signal system were collected, as is shown in Table 1. The degree of saturation at each entrance of every intersection was also captured in real time.

### 4.2. Results

The evacuation problem was simulated using MATLAB R2018b (version 11.4) software (MathWorks, Natick, MA, USA). The parameters are shown in Table 2.

#### 4.2.1. Optimal Evacuation Routes

We obtained an optimal evacuation plan from the outcome of the model, shown in Table 3. During the whole evacuation process, due to the background traffic flows in the road network, different amounts of vehicles were evacuated through the four exits of the parking lot, i.e., 216 vehicles are evacuated through exit E1 to destinations (A1, A2, A3, A17, and A18) along five routes; 264 vehicles were evacuated through exit E2 to destinations (A12, A13, A14, A15, and A16) along five routes; 196 vehicles were evacuated through exit E3 and followed four routes to their destinations (A8, A9, A10, and A11); 184 vehicles were evacuated through exit E4 and followed four routes to the destinations (A4, A5, A6, and A7). Figure 5 reveals the total number of vehicles evacuated along different optimal routes from the four exits. We also know that the vehicles evacuated were inclined to choose the fastest route other than the shortest route.

The standard deviation of the number of vehicles evacuated on each route is 11.8, which means there were differences in the number of vehicles evacuated on each of the 18 routes, i.e., there was an inequality of spatial distribution of vehicle evacuation.

#### 4.2.2. Dynamic Evacuation Process

Vehicle evacuation for a parking lot is a dynamic process, i.e., the vehicle departure rate in a parking lot has a dynamic time-variation, and vehicles choose their optimal routes dynamically because of the dynamic background traffic flows in the road network. Departure curves plotted in Figure 6 display the cumulative vehicles evacuated through the four exits of the parking lot to their destinations along all routes at different times during the evacuation process. 

It was found that, according to the variation of background traffic flows, each exit of the parking lot generated different numbers of vehicles that selected their optimal routes dynamically. For example, all four exits generated 80 vehicles during the first five minutes, but they generated 112 vehicles during the third five minutes. Three vehicles were evacuated along route A13 during the first five minutes, but 11 vehicles were generated during the seventh five minutes.

We also find that it took 42 minutes for exit E1 to clear all 216 vehicles, 40 minutes for exit E2 to clear all 264 vehicles, 41 minutes for exit E3 to clear all 196 vehicles, and 39 minutes for exit E4 to clear all 184 vehicles. Vehicles through exit E2 and E4 had a maximum evacuation time, i.e., 58 minutes. Vehicles through exit E2 had the highest average departure rate, i.e., 6.6 vehicles per minute, while vehicles through exit E4 had the lowest average departure rate, i.e., 4.7 vehicles per minute.

Additionally, we also calculated the traffic flows of different road segments during the evacuation process. Figure 7 indicates the comparison of traffic flows on every road segment with and without evacuation. It was found that the four road segments connecting the four exits of the parking lot had the largest increase in traffic flows during the evacuation process, while five road segments (C4, C9), (C5, C10), (C8, C13), (C2, C3), and (C15, C16) had no change in traffic flows because they did not lie on the optimal routes chosen by vehicles. 

#### 4.2.3. Average Queuing Time and Travel Time

Besides the evacuation time, we also calculated the average queuing time and average travel time of the vehicles evacuated from each of the four exits of the parking lot, as shown in Figure 8. Due to the higher background traffic flows on links (C6, C7) and (C7, C12), the departure rate of vehicles out of exits E3 and E4 was lower, which made these vehicles consume more time in a queuing system, i.e., the average queuing time of queuing systems E3 and E4 was 26.1 minutes and 25.4 minutes, respectively. However, since the evacuation routes from exits E1 and E2 in the buffers had lower background traffic flow volumes, the travel time of vehicles out of these two exits was lower, i.e., the average travel time was 15.8 minutes and 14.7 minutes, respectively.

## 5. Discussion

### 5.1. Optimal Evacuation Strategy

Since it is an important part of disaster mitigation and emergency disposal, an optimal evacuation strategy is usually dependent on road network capacities, traffic management equipment, emergency response resources, real-time traffic conditions, etc. Different from previous research [44,45,46], we divided the evacuation process of a parking lot into two periods. In the first period, a queuing theory was used to estimate the queuing time. In the second period, a traffic flow equilibrium model and an intersection delay model were employed to simulate vehicles’ route choice. Compared to the study of Zheng et al. [47], we not only estimated the queuing time and travel time but also computed the delay at the intersections. As such, we could obtain a more comprehensive and accurate estimation of total evacuation times. Background traffic flow is always less considered in evacuation problems [48], but it actually has a dynamic impact on the two-period evacuation process of a parking lot. It was found that vehicles’ departure rate of every exit is mainly dependent on the time headway of the background traffic flows on the road connecting the exit of the parking lot. We also know that the dynamic distribution of background traffic flows in the evacuation road network influences the optimal evacuation routes greatly; this observation is in line with the study of Alam et al. [49]. Additionally, to our knowledge, the method proposed in this research is an optimization model with minimum evacuation time, which is also applicable in the vehicle evacuation problems of other infrastructures, e.g., tunnels and bridges.

### 5.2. Sensitivity Analysis of $\tau_{r}$

$\tau_{r}$ is the minimum threshold of time headway of the background traffic flows on roads connecting the exits, which allows the vehicles leaving the exits to merge into background traffic flows. Meanwhile, $\tau_{r}$ determines the vehicle departure rate through the exits, and there is a positive relationship between $\tau_{r}$ and the vehicle departure rate. In this section, we analyze the sensitivity of evacuation efficiency relative to the variations of $\tau_{r}$.

We simulated the variations of average queuing time, average travel time, and total evacuation time with different values of $\tau_{r}$, as shown in Figure 9. The higher the $\tau_{r}$ is, the more vehicles can leave the exits during a certain time period. Since a higher $\tau_{r}$ increases the departure rate but also leads to a higher traffic delay on the evacuation routes, there is a negative relationship between average travel time and $\tau_{r}$. While the fluctuation in total evacuation time is an inverted U-shaped curve. $\tau_{r}\;=\;6$ is the minimum point for total evacuation time. When $\tau_{r}\;<\;6$, the average travel time is the main contribution to the total evacuation time, but a lower $\tau_{r}$ has a higher possibility to bring about traffic accidents. The total evacuation time is mainly affected by the average queuing time when $\tau_{r}\;>\;6$. Hence, the optimal value setting of $\tau_{r}$ is a tradeoff between the queuing time and travel time. 

## 6. Conclusions

An optimal evacuation plan for parking lots aims to guide the vehicles from a parking lot following the optimal routes to safe areas with a minimum total evacuation time. In reality, the background traffic flows in a road network always change dynamically over time, and this greatly affects the departure rate of vehicles leaving out of parking lots and determines the optimal evacuation route selection for vehicles. In view of this, this research proposed an optimal evacuation model with total evacuation time minimization using a queuing theory combined with a traffic flow equilibrium model and an intersection delay model. This new model can estimate evacuation time, including queuing time and travel time, and can simulate optimal evacuation routes. A modified ant colony algorithm was developed as the model solution. In addition, a numerical example was employed to test the model. The main results are described as follows.
(1)The simulation results of the numerical example indicate that the optimal evacuation model proposed in this research has an advantage in estimating the total evacuation time comprehensively and can generate a dynamic evacuation plan according to the time variation of background traffic flows in a road network.(2)The results also show that vehicle evacuation for the parking lot is a dynamic process. In other words, the vehicle departure rate in a parking lot has a dynamic time-variation, and the vehicles choose their optimal routes dynamically according to dynamic background traffic flows in the road network.(3)The sensitivity analysis reveals that the minimum threshold of time headway $\tau_{r}$ has a positive relationship with the average queuing time, a negative relationship with the average travel time, and a U-shaped relationship with the total evacuation time, which means that $\tau_{r}$ has a great impact on optimal evacuation plans.

Due to their importance and necessity, optimal evacuation problems have attracted tremendous attention from lots of researchers. However, the interaction between background traffic flows and evacuation strategy has rarely been investigated. This research intends to provide a reference to improve evacuation efficiency. However, there are two limitations of this research. Firstly, we only focused on a parking lot with single-lane exits instead of multi-lane exits. Secondly, some uncertainties, such as weather conditions affecting travel time and the impact of secondary accidents on route choice, were not involved in this research. These limitations will be investigated in a following work.

## Figures and Tables

**Figure 1 ijerph-16-02194-f001:**
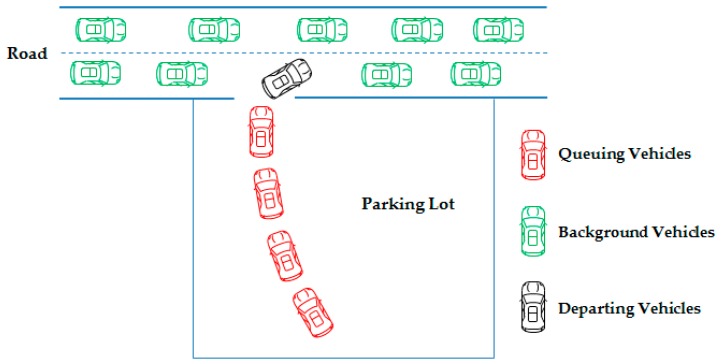
Vehicle evacuation process in a parking lot.

**Figure 2 ijerph-16-02194-f002:**
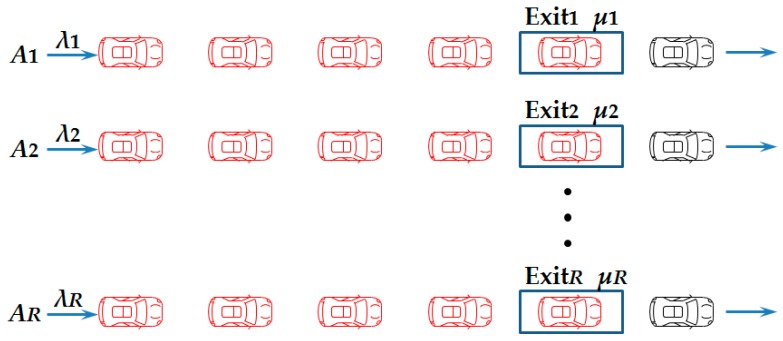
Queuing in a parking lot with *R* exits.

**Figure 3 ijerph-16-02194-f003:**
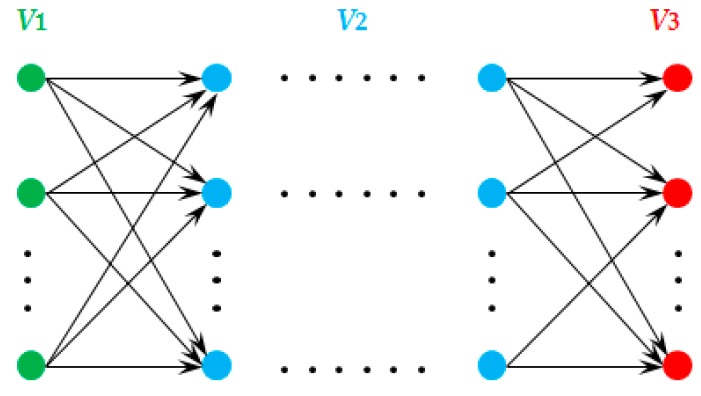
An evacuation network with multi-origin and multi-destination.

**Figure 4 ijerph-16-02194-f004:**
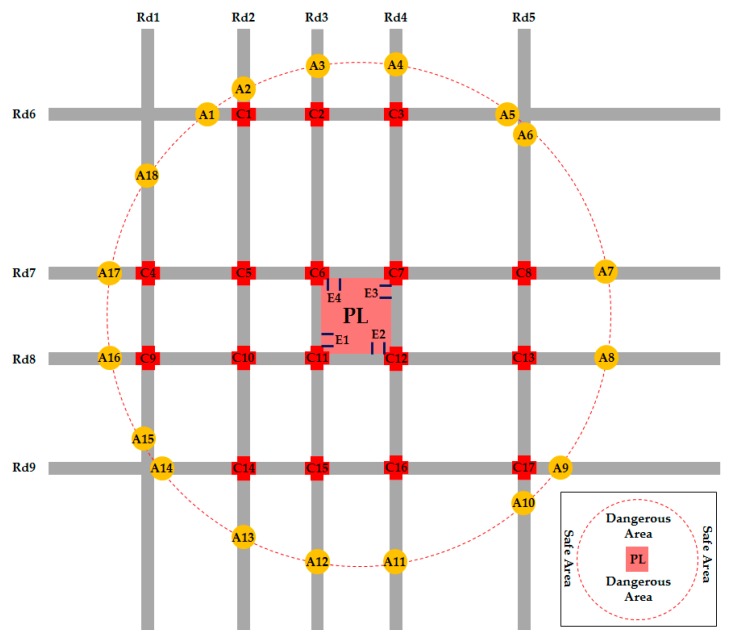
Configuration of the parking lot and road network layout.

**Figure 5 ijerph-16-02194-f005:**
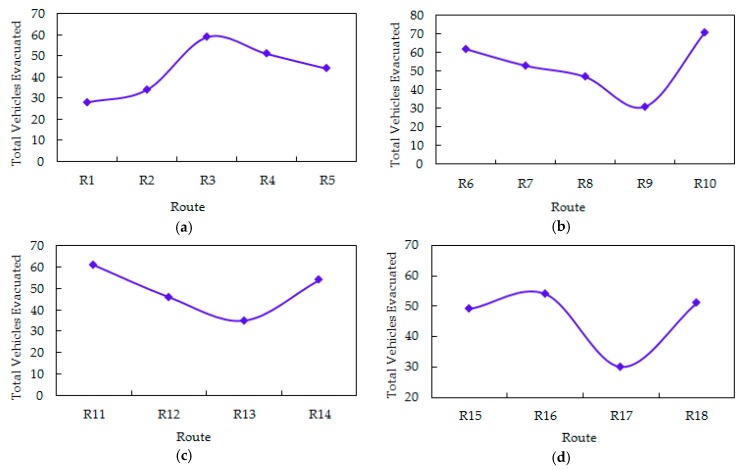
Total vehicles evacuated along different optimal routes from the four exits of the parking lot. (**a**) Exit E1. (**b**) Exit E2. (**c**) Exit E3. (**d**) Exit E4.

**Figure 6 ijerph-16-02194-f006:**
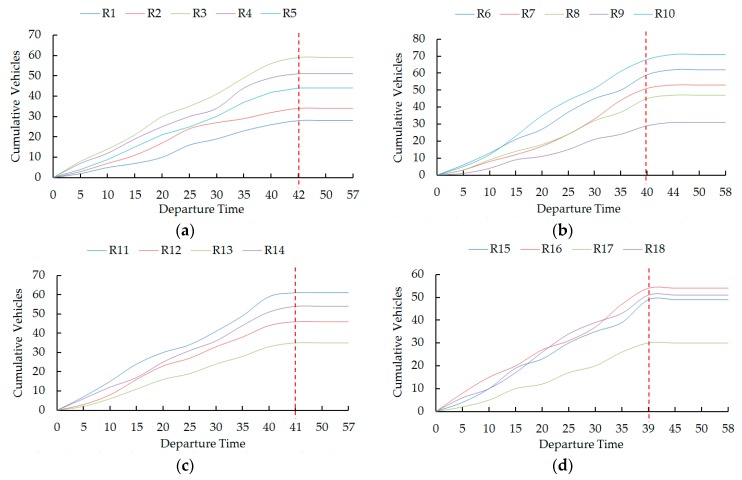
Dynamic evacuation process from the four exits of the parking lot. (**a**) Exit E1. (**b**) Exit E2. (**c**) Exit E3. (**d**) Exit E4.

**Figure 7 ijerph-16-02194-f007:**
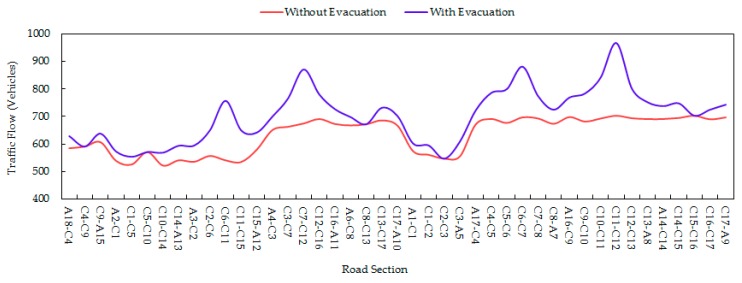
Traffic flows on every road segment with and without evacuation.

**Figure 8 ijerph-16-02194-f008:**
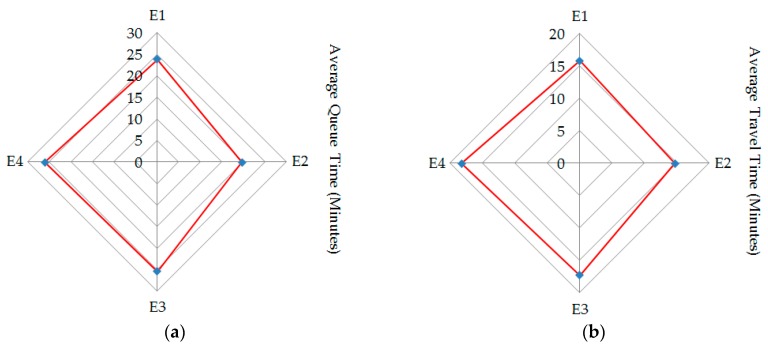
Average queuing time and average travel time of the vehicles evacuated from the four exits of the parking lot. (**a**) Average queuing time. (**b**) Average travel time.

**Figure 9 ijerph-16-02194-f009:**
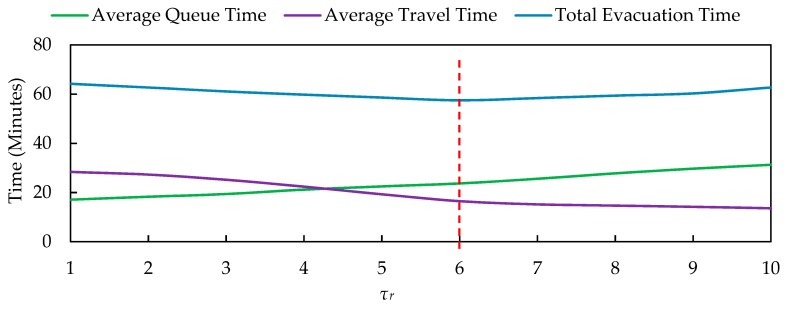
Sensitivity analysis of $\tau_{r}$.

**Table 1 ijerph-16-02194-t001:** Parameters of each traffic signal systems at the intersections.

Intersection	Signal Cycle Time (s)	Green Time Ratio	Intersection	Signal Cycle Time (s)	Green Time Ratio
C1	145	0.42	C10	125	0.45
C2	140	0.42	C11	106	0.46
C3	130	0.48	C12	144	0.41
C4	138	0.50	C13	138	0.43
C5	140	0.41	C14	125	0.46
C6	135	0.45	C15	125	0.46
C7	120	0.45	C16	142	0.37
C8	118	0.48	C17	135	0.34
C9	122	0.50			

**Table 2 ijerph-16-02194-t002:** Parameter settings.

Parameters	Value	Parameters	Value	Parameters	Value
$\mu_{r}$	3 s	ξ	2.82	m	30
$\tau_{r}$	6 s	ω	0.65	α	0.5
$v_{ij0}$	80 km/h	$N_{\mathrm{iteration}}$	70	β	0.5
ψ	0.48	$\vartheta_{ij}\left(0\right)$	20	ρ	0.2

**Table 3 ijerph-16-02194-t003:** Optimal evacuation routes.

Original Node	Destination Node	Route	Route ID	Total Vehicles Evacuated	Length (km)
E1	A1	E1-C6-C5-C1-A1	R1	28	4.8
	A2	E1-C6-C2-C1-A2	R2	34	4.7
	A3	E1-C6-C2-A3	R3	59	4.4
	A17	E1-C6-C5-C4-A17	R4	51	3.3
	A18	E1-C6-C5-C4-A18	R5	44	4.3
E2	A12	E2-C11-C15-A12	R6	62	4.4
	A13	E2-C11-C15-C14-A13	R7	53	4.5
	A14	E2-C11-C10-C14-A14	R8	47	4.4
	A15	E2-C11-C10-C9-A15	R9	31	3.9
	A16	E2-C11-C10-C9-A16	R10	71	3
E3	A8	E3-C12-C13-A8	R11	61	4.2
	A9	E3-C12-C13-C17-A9	R12	46	5.1
	A10	E3-C12-C16-C17-A10	R13	35	4.9
	A11	E3-C12-C16-A11	R14	54	4.3
E4	A4	E4-C7-C3-A4	R15	49	4.1
	A5	E4-C7-C3-A5	R16	54	5.4
	A6	E4-C7-C8-A6	R17	30	5.3
	A7	E4-C7-C8-A7	R18	51	4.1

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
