# Peer review of "Optimal Evacuation Strategy for Parking Lots Considering the Dynamic Background Traffic Flows"

_ijerph, 2019, doi:10.3390/ijerph16122194_

Round 1
Reviewer 1 Report
The paper entitled "Optimal Evacuation Strategy for the Parking Lot Considering the Dynamic Background Traffic Flows" introduces a novel methodology for a situation-aware evacuation strategy. The authors explain in a broad introduction and state-of-the-art section the manyfold component respected in their approach. The references are numerous and suitable for this paper. Subsequently, the modelling part is understandable due to the well-designed structure and clear depictions in the figures. Section 4 summerizes the procedure with an example. Section 5 and 6, discussion and conclusions are compact and appropriate. Nevertheless, the statement in line 288 is unnecessary, since coding model and algorithm is possible with any other software.
All in all, a substantial paper that can find its readers.
Author Response
Point 1: The paper entitled "Optimal Evacuation Strategy for the Parking Lot Considering the Dynamic Background Traffic Flows" introduces a novel methodology for a situation-aware evacuation strategy. The authors explain in a broad introduction and state-of-the-art section the manyfold component respected in their approach. The references are numerous and suitable for this paper. Subsequently, the modelling part is understandable due to the well-designed structure and clear depictions in the figures. Section 4 summerizes the procedure with an example. Section 5 and 6, discussion and conclusions are compact and appropriate.
Response 1: The authors greatly appreciate the reviewer’s encouragement and suggestions. The authors have revised the manuscript according to the reviewer’s comments.
Point 2: Nevertheless, the statement in line 288 is unnecessary, since coding model and algorithm is possible with any other software.
Response 2: The authors have revised the statement as “The evacuation problem was simulated in the MATLAB environment”. Please refer to Line 302.

Reviewer 2 Report
The article deals with an important issue from the point of view of road safety and studies the case related to the emergency evacuation from a parking facility. For this reason, I recommend the publication of this manuscript in International Journal of Environmental Research and Public Health.
As recommendations for improvement, I suggest the following points:
- do not include results in the introduction, as in the second paragraph.
- revise some expressions with the “can” verb, they may be wrong.
Author Response
Point 1: The article deals with an important issue from the point of view of road safety and studies the case related to the emergency evacuation from a parking facility. For this reason, I recommend the publication of this manuscript in International Journal of Environmental Research and Public Health. As recommendations for improvement, I suggest the following points.
Response 1: The authors greatly appreciate the reviewer’s encouragement and suggestions. The authors have revised the manuscript according to the reviewer’s comments.
Point 2: do not include results in the introduction, as in the second paragraph.
Response 2: The authors have moved the second paragraph to Sectioin 3.2. Please refer to Lines 130-Lines 145.
Point 3: revise some expressions with the “can” verb, they may be wrong.
Response 3: The authors have revised some expressions with the “can” verb as follows:
Line 77: revise “which can be classified into ...” as “which are classified into ...”
Line 89: revise “and can make the best of” as “and make the best of”
Line 125: revise “which can only serve one vehicle” as “which only serves one vehicle”
Line 129: revise “Vehicles evacuated from the parking lot can only turn right into the road network” as “Vehicles evacuated from the parking lot enter into the road network only by right turn”
Line 156: revise “r can be calculated as” as “r is calculated as”
Line 169-170: revise “which can be formulated as” as “which is formulated as”
Line 174: revise “traffic flow can be calculated as” as “traffic flow is calculated as”
Line 181: revise “vehicles’ queuing time in the parking lot can be calculated as” as “vehicles’ queuing time in the parking lot is calculated as”
Line 217: revise “which can be formulated as” as “which is formulated as”
Line 223: revise “traffic equilibrium can be formulated as” as “traffic equilibrium is formulated as”
Line 231: revise “vehicles evacuated can be calculated as” as “vehicles evacuated is calculated as”
Line237: revise “can be calculated as” as “are calculated as”
Line 277: revise “can be calculated by Equation (30)” as “is calculated by Equation (30)”
Line 287: revise “V1={E1,E2,E3,E4}can be denoted as” as “V1={E1,E2,E3,E4} is denoted as”
Line 386: revise “τr can increase the departure rate” as “τr increases the departure rate”

Reviewer 3 Report
The authors present a novel method to optimize the evacuation of a parking lot, considering dynamic background traffic flow in nearby streets. For that, they use a two-step methodology, considering queuing theory for estimating parking lot exit delay, traffic flow equilibrium and intersection delay models for simulating route choice and a modified ant colony algorithm. The methodology is subsequently applied into a case study based on an actual parking lot in Xi'an (China) with 860 vehicles inside.
The manuscript has a correct and clear structure, and the topic could be of some interest for IJERPH journal readers. However, in opinion of this reviewer, the authors should clarify some concerns and questions before it can be published.
First of all, a grammar revision, preferably by a professional proofreading service, is strongly advised. There are several grammar mistakes and typos that should be amended before the publication in this journal.
Section 4.1: The authors should expand this subsection including some tables and/or figures with the traffic parameters and data of the simulated road network (length, traffic capacity, free flow speed, intersection signal cycles, traffic volumes, etc.) in order to the model could be replicated in full by other researchers.
Section 4.1.2: It should be clearer for the reader to include a new figure showing the different optimal routes for each parking lot exit (one drawing for each exit)
Section 4.2.2: In order to put into context the results obtained, a figure showing the mean flow of different road sections could be very helpful.
Section 5 (Discussion) is very short and only focuses on sensitivity analysis. The authors should also discuss the results previously shown and contrast them with existing evacuation models proposed by other authors. Maybe they can expand here the route analysis made in Section 4.2.2 too.
A Nomenclature appendix is also strongly recommended.
Specific points:
L144, 158, 218, 221 and wherever else: Delete the word "Reference" before the number
L182-201: The initial "The" is unnecessary
Author Response
Point 1: The authors present a novel method to optimize the evacuation of a parking lot, considering dynamic background traffic flow in nearby streets. For that, they use a two-step methodology, considering queuing theory for estimating parking lot exit delay, traffic flow equilibrium and intersection delay models for simulating route choice and a modified ant colony algorithm. The methodology is subsequently applied into a case study based on an actual parking lot in Xi'an (China) with 860 vehicles inside. The manuscript has a correct and clear structure, and the topic could be of some interest for IJERPH journal readers. However, in opinion of this reviewer, the authors should clarify some concerns and questions before it can be published.
Response 1: The authors greatly appreciate the reviewer’s encouragement and suggestions. This memo documents our responses to all review comments. The appropriate changes have been made to the manuscript.
Point 2: First of all, a grammar revision, preferably by a professional proofreading service, is strongly advised. There are several grammar mistakes and typos that should be amended before the publication in this journal.
Response 2: The authors have got a professional proofreading service, and revised the grammar mistakes and typos as follows:
Line 42: revise “more than one exits” as “more than one exit”
Line 44: revise “which is dependent on” as “which are dependent on”
Line 60: revise “including queuing time, travel time” as “including the queuing time, the travel time”
Line 87: revise “road network based approach” as “road network-based approach”
Line 186: revise “muti-origin -destination” as “multi-origin and multi-destination”
Line 249: revise “a common method for solving optimization problem” as “a common method to solve optimization problem”
Point 3: Section 4.1: The authors should expand this subsection including some tables and/or figures with the traffic parameters and data of the simulated road network (length, traffic capacity, free flow speed, intersection signal cycles, Traffic Volumes, etc.) in order to the model could be replicated in full by other researchers.
Response 3: The authors have added a table, which gives the parameters of each traffic signal systems at the intersections, please refer to Line 300, Table 1. The authors also added a table to provide the data of the road network and traffic volumes, please refer to Line483-Line 486, Appendix B, Table B1,
Point 4: Section 4.2.1: It should be clearer for the reader to include a new figure showing the different optimal routes for each parking lot exit (one drawing for each exit).
Response 4: The authors have added a figure in Section 4.2.1, which shows the total vehicles evacuated along different optimal routes from the 4 exits of the parking lot. Please refer to Figure 5.
Point 5: Section 4.2.2: In order to put into context the results obtained, a figure showing the mean flow of different road sections could be very helpful.
Response 5: The authors have added a figure in Section 4.2.2, which shows the mean flow of different road sections. Please refer to Figure 7. And traffic flow analysis of Figure 7 is also made. Please refer to Line 341-Line 346.
Point 6: Section 5 (Discussion) is very short and only focuses on sensitivity analysis. The authors should also discuss the results previously shown and contrast them with existing evacuation models proposed by other authors. Maybe they can expand here the route analysis made in Section 4.2.2 too.
Response 6: The authors have added the discussion of the key findings of this research to Section 5 (Discussion). Please refer to Line 360-Line 377, Section 5.1.
Point 7: A Nomenclature appendix is also strongly recommended.
Response 7: The authors have added a Nomenclature appendix. Please refer to Line 436-Line 481, Appendix A.
Point 8: L144, 158, 218, 221 and wherever else: Delete the word "Reference" before the number.
Response 8: The authors have removed the word "Reference". Please refer to Line 156, Line 170, Line 229 and Line 233.
Point 9: L182-201: The initial "The" is unnecessary.
Response 9: The authors have removed the initial "The". Please refer to Lines 194-213.
Round 2
Reviewer 3 Report
The authors have addressed satisfactorily most of this reviewer's questions and suggestions. However, in opinion of this reviewer, some minor changes are still required before its final publication.
The notation list (L189-212) contains duplicated items with the one in Appendix A, so it better should be deleted and merged with the one displayed in Appendix A.
Please, use two decimal places in all the green time ratios shown in Table 1.
Section codes (1 to 43) should be added to Figure 4 or, alternatively, the authors should use node-to-node notation in Figure 7, in order to be easily understood.
Please, correct vertical axis units in Figure 7: (vehicles) instead of (vehicle).
Author Response
Point 1: The authors have addressed satisfactorily most of this reviewer's questions and suggestions. However, in opinion of this reviewer, some minor changes are still required before its final publication.
Response 1: The authors greatly appreciate the reviewer’s encouragement and suggestions. The authors have revised the manuscript according to the reviewer’s comments.
Point 2: The notation list (L189-212) contains duplicated items with the one in Appendix A, so it better should be deleted and merged with the one displayed in Appendix A.
Response 2: The authors have removed the notation list. All the notations was displayed in Appendix A.
Point 3: Please, use two decimal places in all the green time ratios shown in Table 1.
Response 3: The authors have changed all the green time ratios into two decimal places. Please refer to Table 1.
Point 4: Section codes (1 to 43) should be added to Figure 4 or, alternatively, the authors should use node-to-node notation in Figure 7, in order to be easily understood.
Response 4: The authors have changed the notation of X-axis in Figure 7 into node-to-node notation. Please refer to Figure 7.
Point 5: Please, correct vertical axis units in Figure 7: (vehicles) instead of (vehicle).
Response 5: The authors have revised the vertical axis unit as “vehicles”. Please refer to Figure 7.
